# Functional and Molecular Characterisation of Heart Failure Progression in Mice and the Role of Myosin Regulatory Light Chains in the Recovery of Cardiac Muscle Function

**DOI:** 10.3390/ijms23010088

**Published:** 2021-12-22

**Authors:** Kasturi Markandran, Haiyang Yu, Weihua Song, Do Thuy Uyen Ha Lam, Mufeeda Changaramvally Madathummal, Michael A. Ferenczi

**Affiliations:** 1Lee Kong Chian School of Medicine, Nanyang Technological University, Experimental Medicine Building, 59 Nanyang Drive, Singapore 636921, Singapore; kast0008@e.ntu.edu.sg (K.M.); yhy0601@163.com (H.Y.); weihua.song@innolandbio.com (W.S.); lam_do_thuy_uyen_ha@gis.a-star.edu.sg (D.T.U.H.L.); mufeeda001@e.ntu.edu.sg (M.C.M.); 2Laboratory of Precision Disease Therapeutics, Genome Institute of Singapore, 60 Biopolis Street, Singapore 138672, Singapore; 3Department of Medicine, Yong Loo Lin School of Medicine, National University of Singapore, 10 Medical Drive, Singapore 117597, Singapore; 4A*STAR Microscopy Platform—Electron Microscopy, 61 Biopolis Drive, Proteos, Singapore 138673, Singapore; 5Brunel Medical School, Brunel University London, Kingston Lane, Uxbridge UB8 3PH, UK

**Keywords:** cardiac regulatory light chain, enzymes, heart failure, isometric force, phosphorylation

## Abstract

Heart failure (HF) as a result of myocardial infarction (MI) is a major cause of fatality worldwide. However, the cause of cardiac dysfunction succeeding MI has not been elucidated at a sarcomeric level. Thus, studying the alterations within the sarcomere is necessary to gain insights on the fundamental mechansims leading to HF and potentially uncover appropriate therapeutic targets. Since existing research portrays regulatory light chains (RLC) to be mediators of cardiac muscle contraction in both human and animal models, its role was further explored In this study, a detailed characterisation of the physiological changes (i.e., isometric force, calcium sensitivity and sarcomeric protein phosphorylation) was assessed in an MI mouse model, between 2D (2 days) and 28D post-MI, and the changes were related to the phosphorylation status of RLCs. MI mouse models were created via complete ligation of left anterior descending (LAD) coronary artery. Left ventricular (LV) papillary muscles were isolated and permeabilised for isometric force and Ca^2+^ sensitivity measurement, while the LV myocardium was used to assay sarcomeric proteins’ (RLC, troponin I (TnI) and myosin binding protein-C (MyBP-C)) phosphorylation levels and enzyme (myosin light chain kinase (MLCK), zipper interacting protein kinase (ZIPK) and myosin phosphatase target subunit 2 (MYPT2)) expression levels. Finally, the potential for improving the contractility of diseased cardiac papillary fibres via the enhancement of RLC phosphorylation levels was investigated by employing RLC exchange methods, in vitro. RLC phosphorylation and isometric force potentiation were enhanced in the compensatory phase and decreased in the decompensatory phase of HF failure progression, respectively. There was no significant time-lag between the changes in RLC phosphorylation and isometric force during HF progression, suggesting that changes in RLC phosphorylation immediately affect force generation. Additionally, the in vitro increase in RLC phosphorylation levels in 14D post-MI muscle segments (decompensatory stage) enhanced its force of isometric contraction, substantiating its potential in HF treatment. Longitudinal observation unveils potential mechanisms involving MyBP-C and key enzymes regulating RLC phosphorylation, such as MLCK and MYPT2 (subunit of MLCP), during HF progression. This study primarily demonstrates that RLC phosphorylation is a key sarcomeric protein modification modulating cardiac function. This substantiates the possibility of using RLCs and their associated enzymes to treat HF.

## 1. Introduction

Heart failure (HF) is a clinical syndrome where the ability of the heart to pump blood to the rest of the body is compromised. HF arises either alone or as a combination of dysfunctional pericardium, myocardium, endocardium, valves or major vessels [1]. Patients typically present either HF with preserved ejection fraction (HFpEF) or reduced ejection fraction (HFrEF). As these are merely classifications of HF, HFpEF can progress into HFrEF as the condition deteriorates. HFrEF is a more serious condition with a low ejection fraction of <45% [1]. In general, the prevalence of HF is increasing globally, due to an ageing population. About 64.3 million people are affected by it worldwide and 4.5% of them are Singaporeans, while 3–4% of them are Europeans and Americans [2,3]. Recently, increasing HF incidence in youths has been observed, potentially due to increases in obesity. If this trend continues, the burden of HF is expected to increase [4].

The most common cause of HF is myocardial infarction (MI), which occurs as a result of severe localised ischaemia due to blockage in the coronary artery [5]. Following MI, the left ventricle (LV), which is primarily responsible for pumping blood throughout the body, undergoes substantial remodelling (i.e., hypertrophy, dilation, changes in contraction characteristics, post-translational modification of various proteins, etc.) as a compensatory effect to sustain cardiac functionality [5,6,7,8]. However, these compensatory mechanisms become physiologically ineffective in the long term, causing the heart to slip into the decompensatory state [9]. The mechanisms behind the transition from the compensatory to the decompensatory state are unclear. Given the importance of sarcomeric proteins in muscle contraction, it is hypothesied that alterations to these proteins are the root cause of this transition.

Aligned with the above hypothesis, a study using an MI rat model showed that the contractile twitch force of the cardiac trabeculae correlates with the myosin heavy chain content in end-stage HF [10], while the power output by myocardial regions unaffected by the MI is reduced in human HF [11]. Another study by Toepfer et al., showed that phosphorylation levels of sarcomeric proteins vary at compensatory and decompensatory stages of HF progression [12]. These studies highlight the fact that sarcomeric protein modifications can potentially induce cardiac functional changes during disease progression. Thus, designing drugs targeting myofilament proteins may be a viable therapeutic approach to alleviate the primary defects in the sarcomere under diseased state conditions [13].

In this study, we focused on myosin regulatory light chains (RLCs), as they play an important role in cardiac physiology regulation [14,15]. RLC phosphorylation facilitates force production by increasing the maximal isometric tension, calcium sensitivity of force and rate of force redevelopment at intermediate and maximal calcium activation levels [16,17], and also enhances the motility of human beta-myosin under load as seen by in vitro motility assays [18]. The levels of RLC phosphorylation decreased in end-stage HF of both human and animals [19]. However, it is not known whether the changes in RLC phosphorylation drive the functional changes during HF progression, or vice versa.

A study reported by Stull et al., showed that RLC phosphorylation potentially inhibits pathological cardiac hypertrophy [20]. Other studies have shown that enhancement of RLC phosphorylation in cardiac fibres increases the basal force of contraction or rescues the pathological effects arising from mutated RLCs [11,21]. This raises the question of whether the manipulation of RLC phosphorylation levels at a cellular level could be an effective therapeutic strategy to reverse contractile defects during HF progression. At the moment, there are no drugs that target RLCs directly. Propanolol was suggested to regulate RLC phosphorylation via the inhibition of the β-adrenergic pathway. However, it was ineffective, suggesting that the β-adrenergic pathway does not primarily influence RLC phosphorylation [15].

In this study, we used an MI mouse model to study the time course of changes in maximal isometric force, calcium sensitivity (EC50) and RLC phosphorylation modifications between 2D and 28D post-MI. We also manipulated RLC phosphorylation levels in permeabilized diseased papillary fibres by exchanging native RLCs with phosphorylated recombinant RLCs and measured the corresponding muscle contraction to observe their recovery. Finally, to gain mechanistic insights, we quantified the expression levels of key enzymes (MLCK, ZIPK, MYPT2) facilitating cardiac RLC and other regulatory sarcomeric proteins’ (TnI, MyBP-C) phosphorylation levels during HF progression.

When the longitudinal profiles of maximal isometric force and RLC phosphorylation were superimposed, they indicate that changes in RLC phosphorylation immediately affect force generation, in a way that promotes contractility of diseased hearts. Additionally, we showed that enhancing RLC phosphorylation levels reverses compromised contractility in decompensated papillary muscle segments. The longitudinal enzyme quantification unveils the possibility of enzymes such as MYPT2 (other than MLCK and ZIPK) being involved in regulating RLC phosphorylation levels during HF progression. The quantification of sarcomeric protein phosphorylation and enzyme levels suggests that the resultant cardiac functionality is a complex interplay of varying sarcomeric proteins. Bringing these results together, we link mechanical, biochemical and physiological changes in the heart tissue during the progression of HF. These results demonstrate that RLC phosphorylation is a key sarcomeric protein modification that influences cardiac function during HF progression and that RLCs and their associated enzymes have potential in the treatment and management of HF.

## 2. Results

### 2.1. Myocardial Infarction (MI) Mouse Model Appropriate for HF Progression Study

The ligature position was optimised by experimenting at two positions, just above the bifurcation LAD (H) or just below the bifurcation (LAD (L)) (Figure 1A). The longitudinal contractile forces at saturating calcium concentration for the LAD (L), LAD (H) and the control group were also monitored (Figure 1C). At 4 weeks post-MI, the contractile force was 4.34 ± 0.3 kN/m^2^ when ligated at LAD (H) and 7.63 ± 0.3 kN/m^2^ when ligated at LAD (L). Similarly, the force at 8 weeks post-MI was lower when ligated at LAD (H) than LAD (L). This shows that the rate of functional deterioration is higher when ligated at LAD (H) as compared to LAD (L). Since we aim to study HF progression over a short period of 28 days, ligating at LAD (H) will be extensive and appropriate to capture the advancement. Thus, this position was ligated to create an MI mouse model throughout the study.

To confirm that open chest surgery and ageing during the time course of the experiments do not affect cardiac muscle contractile force, the contractile force of the permeabilised papillary muscle in two control groups (age-matched control and age-matched sham surgery groups) was studied (Figure 1 and Appendix A). The average contractile force of permeabilised papillary muscle in the control group was 8.15 ± 2.06 kN/m^2^ and the average contraction force of the sham group was 7.11 ± 0.83 kN/m^2^. There was no significant difference between these two groups across the 28 day experimental period. Thus, we concluded that the changes that occurred following LAD ligation were not a consequence of ageing, open chest surgery or the recovery from it.

To validate HF progression, key physiological and biological markers such as hypertrophic indices and brain natriuretic peptide (BNP) expression levels were monitored. Hypertrophy is the thickening of the heart due to the enlargement of cells and other alterations, such as extracellular matrix deposition [22]. Thus, the degree of hypertrophy is primarily determined by the increase in heart mass. The heart mass is normalised by the tibia length (which is consistent among populations of the same age group). The hypertrophic index (heart weight/tibia length ratio) was generally higher in post-MI groups than sham groups. However, the differences were statistically significant between the post-MI and sham groups in 10D, 14D and 28D post-MI groups (Figure 2A). This is clearly represented in the relative hypertrophic indices in Figure 2B.

The brain natriuretic peptide (BNP) is a cardiac neurohormone biomarker predominantly secreted by the ventricular myocardium in response to increased wall stress [23,24,25,26]. BNP is one of the gold standard techniques [27,28], alongside echocardiography, for diagnosis, stratifying risks and making therapeutic decisions for HF [23]. BNP expression levels are low in age-matched control (0.102 ± 0.07). There was an increase in both 7D (0.315 ± 0.07) and 14D post-MI (0.164 ± 0.04) when compared to their respective age-matched control (0.102 ± 0.07) (Figure 2C). The fold change also showed that the BNP levels at 7D post-MI were higher as compared to 14D post-MI, although statistically insignificant (Figure 2D). These results were aligned with the signs of progressively failing human hearts, where the BNP levels are higher in hypertrophic cardiomyopathy as compared to dilated cardiomyopathy and showing very low levels in control groups [29].

Besides hypertrophic index and BNP gene expression levels, HF progression was evaluated on a structural front. The event of hypertrophy is evident from the obvious ventricular septal hypertrophy at 7D and 14D post-MI in histology sections (Figure 2E) [19]. On the other hand, the heart is significantly enlarged at 14D post-MI as compared to 7D post-MI and the age-matched control (Figure 2E). These are clinical features of dilated cardiomyopathy [20], which eventually lead to HF.

On a microstructural front, the accumulation of inflammatory molecules (in between cardiac muscle fibres) is evident at both 7D and 14D post-MI, as highlighted by red arrows (Figure 2F). Granulation tissues formed as a result of dead cell removal and fibroblast and endothelial cell deposition [21] increased gradually from 7D to 14D post-MI, as indicated by yellow arrows (Figure 2F). Some regions in post-MI samples appeared to be pale (although inconsistent), possibly due to the disruption of sarcolemma and accumulation of water in cardiomyocytes [11].

The results collectively showed that the hearts concurrently displayed characteristics of hypertrophic and dilated cardiomyopathy and that the hearts failed with time (with dilation dominating in the later stages).

### 2.2. Ca^2+^ Sensitivity and Calcium-Induced Cooperativity May Not Be Key Regulators of Force Development during HF Progression

The Ca^2+^ sensitivity and cooperativity of myofilaments contribute to the characteristics of cardiac contraction. EC_50_ is the calcium concentration at which half-maximal force is achieved, which indicates calcium sensitivity of the contractile machinery. A lower EC_50_ indicates an increased Ca^2+^ sensitivity, as a given steady-state isometric force is reached at a lower concentration of free calcium [30]. The Hill coefficient is the slope of the line relating force and the negative logarithm of the calcium concentration. The steeper the slope, the greater the cooperative activation of the sarcomeric contractile elements. This could be such that binding of calcium to one troponin subunit increases Ca^2+^ affinity in the neighbouring troponin on the same actin filament [31].

At 2D and 4D post-MI, the EC_50_ decreased by about 28% when compared to its corresponding sham group, suggesting enhanced calcium sensitivity (Table 1 and Figure 3A). From 7D post-MI onwards, the EC_50_ (half maximal effective concentration) goes back to similar levels as the sham group, as shown in Figure 3. EC_50_ was also lower in the post-MI group compared with sham at 14D and 28D post-MI. However, there is no statistical difference among these times, suggesting that calcium sensitivity may not be the main factor influencing isometric force changes. In Figure 3B, the Hill slope shows enhanced levels at 2D and 4D post-MI, and then decreases between 7D and 10D post-MI and increases slightly at 14D and 28D post-MI. However, there are no statistical differences among these timepoints, suggesting that cooperativity does not primarily influence isometric force changes. However, it is interesting that the data collectively indicate that compensation at 4D post-MI comes with a higher calcium sensitivity and stronger calcium cooperativity.

### 2.3. RLC Phosphorylation Transiently Changes to Sustain Cardiac Function during Compensatory Phase of Heart Failure

The RLC phosphorylation level was lower at 2D post-MI (0.22 ± 0.01) when compared to sham control (0.37 ± 0.02) (Figure 4Aii). The levels increased to higher than those of sham control at 4D (0.35 ± 0.02) and 7D post-MI (0.42 ± 0.02). The phosphorylation levels decreased below those of sham control (0.35 ± 0.01) from 10D post-MI (0.27 ± 0.04) onwards. Similarly, the maximal isometric force decreased at 2D post-MI (5.06 ± 0.3) as compared to its sham control (7.34 ± 0.3) and increased at 4D post-MI (11.0 ± 0.7) (Figure 4Di). At 7D post-MI (7.28 ± 0.7), the force levels were equivalent to those of the sham control (7.31 ± 0.4). The levels decreased below the sham control (7.04 ± 0.3) from 10D post-MI (6.38 ± 1.0) onwards. Based on the isometric force profile and structural and biological markers for MI model verification (with reference to Section 2.1), 2D to 7D post-MI are considered to be in the compensatory phase, while 10D post-MI onwards is considered to be the decompensatory phase. The relative values (post-MI values normalised by average of sham values) of RLC phosphorylation and isometric force levels are presented in Appendix A. It is important to elucidate whether the changes in RLC phosphorylation levels drive functional changes (such as maximal isometric force) during HF progression, or whether HF progression drives changes in RLC phosphorylation status. Thus, the sequence of events of both parameters was superimposed. However, there is no significant time lag observed between the two events (Figure 4E), suggesting that RLC phosphorylation levels change transiently to enhance muscle contraction.

It was also observed that when the phosphorylation levels were between 0.35 and 0.42 mol Pi/mol RLC, the isometric force was sustained at physiologically appropriate levels (Figure 4Aii,Di). This range is close to the baseline RLC phosphorylation levels of ~0.3–0.4 mol Pi/mol RLC, suggesting that maintaining RLC phosphorylation levels to baseline levels could recover cardiac contractility. To test this, the RLC phosphorylation levels in diseased cardiac fibres were manipulated in vitro and the corresponding maximal isometric force was measured. This is discussed in Section 2.5.

### 2.4. Post-Translational Modifications in Left Ventricular Tissue during HF Progression

At 2D post-MI, both RLC and TnI phosphorylation decreased significantly below sham control levels with maximal isometric force, while phosphorylation levels of MyBP-C increased above sham levels (Figure 4E). During what is defined as the compensatory phase, which is between 4D and 7D post-MI, RLC phosphorylation levels were above sham control levels, MyBP-C phosphorylation levels were below sham control levels and TnI phosphorylation levels were maintained at sham control levels (Figure 4E). The transition from compensatory to decompensatory stage occurred around 10D post-MI. After this, the phosphorylation levels of MyBP-C increased above sham control levels, while RLC phosphorylation and isometric force levels decreased below sham control levels until 28D post-MI. TnI was maintained at sham control levels until 28D post-MI, other than a drop at 14D post-MI (Figure 4E).

**Figure 4 ijms-23-00088-f004:**
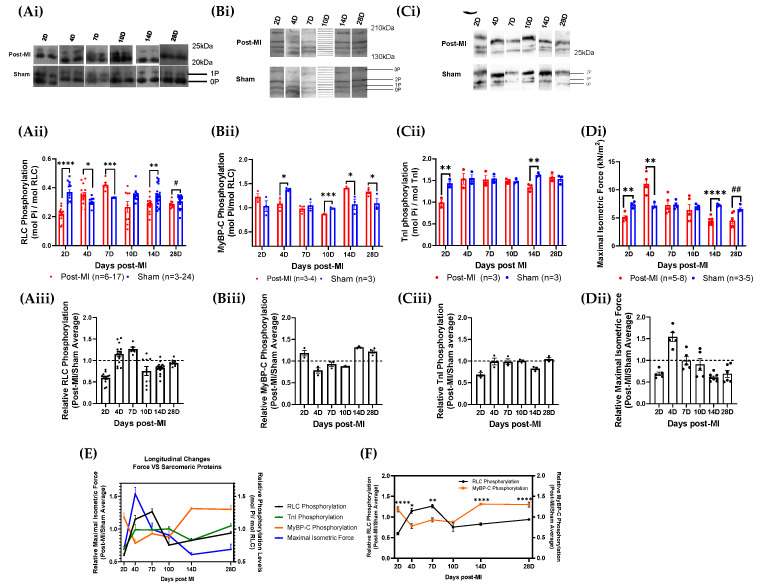
Longitudinal sarcomeric protein phosphorylation and maximal isometric force measurements during HF progression. (**Ai**,**Bi**,**Ci**) Blot images of RLC, MyBP-C and TnI phosphorylation levels of post-MI and sham groups at different time points ranging between 2D and 28D post-MI, respectively. (**Aii**,**Bii**,**Cii**,**Di**) Bar graphs of RLC, MyBP-C and TnI phosphorylation and maximal isometric force of post-MI and sham groups at different time points ranging between 2D and 28D post-MI. Unpaired one-tailed *t*-test (*) was conducted on parametric and homogenous population. Unpaired one-tailed t-test with Welch’s correction (*) was conducted on parametric but inhomogeneous population. Unpaired one-tailed Mann–Whitney test (^#^) was conducted on non-parametric population. Number of mice used is summarised in Appendix A. Note: ‘*n*’ refers to the number of technical replicates. (**Aiii**,**Biii**,**Ciii**,**Dii**) Bar graphs for relative RLC, MyBP-C and TnI phosphorylation and maximal isometric force values during HF progression, respectively. Kruskal-Wallis ANOVA test was performed on RLC phosphorylation populations and showed that the medians varied significantly (* *p* = 0.0102). Ordinary one-way ANOVA performed on MyBP-C phosphorylation showed significant differences in the means (**** *p* < 0.0001). Ordinary one-way ANOVA performed on TnI phosphorylation showed significant differences in the means (** *p* < 0.01). Ordinary one-way ANOVA performed on isometric force data showed significant difference among population means (*p* < 0.0001). (**E**) Line graph of relative phosphorylation levels of all three sarcomeric proteins and isometric force. (**F**) Line graph of relative phosphorylation levels of RLC and MyBP-C. Note: * *p* < 0.05, ** *p* < 0.01, *** *p* ≤ 0.001, **** *p* < 0.0001, ^#^ *p* < 0.05, ^##^ *p* < 0.01.

Thus, the results show that the MyBP-C phosphorylation level decreases in the compensatory phase and increases in the decompensatory phase. Contrary to this, the RLC phosphorylation level increases in the compensatory phase and decreases in the decompensatory phase. TnI, on the other hand, shows no particular change in TnI phosphorylation levels. The results suggest an interrelationship between RLC and MyBP-C phosphorylation (Figure 4F), although its underlying mechanisms are unknown. The longitudinal profile also shows that the MyBP-C phosphorylation pattern presents a progression that is opposite to the isometric force profile (Figure 4E), which is contrary to the current observations. The protein phosphorylation levels are summarised in Appendix A.

The correlation between the changes of the respective sarcomeric protein phosphorylation levels and isometric force was analysed using Spearman’s correlation test. There is a strong and negative correlation between MyBP-C phosphorylation and isometric force (r = −0.94), which is statistically significant (Appendix A).

### 2.5. Modulation of the Phosphorylation Level of RLC Induces Recovery of the Contractile Performance of Papillary Muscle In Vitro

To explore the potential of RLC phosphorylation in reversing diseased muscle contractility, phosphorylated recombinant RLCs were exchanged (in vitro) into permeabilised papillary muscles isolated from 14D post-MI mice hearts. After the exchange procedure, the phosphorylation levels in 14D post-MI fibres were enhanced from 0.09 ± 0.04 mol Pi/mol RLC to 0.33 ± 0.007 mol Pi/mol RLC (Figure 5Aii). However, it was not possible to enhance the phosphorylation levels to match 14D sham (non-exchanged) papillary segments. The maximal isometric force of 14D post-MI fibres was increased from 3.95 ± 0.2 kN/m^2^ to 6.20 ± 0.2 kN/m^2^ after the exchange procedure (with 25 mM P-RLC, 27 mM TnC). However, the force produced did not recover to match the force produced by 14D sham papillary (non-exchanged) segments (Figure 5B). This could be due to RLC phosphorylation levels not being enhanced to suitable levels (0.35–0.42 mol Pi/mol RLC), as mentioned in Section 2.3. Nonetheless, in permeabilised muscle models where other signalling mechanisms are disrupted, increased RLC phosphorylation levels are able to enhance the isometric force of contraction. The phosphorylation and force values are summarised in Appendix A.

### 2.6. Enzyme Quantification of Left Ventricular Tissue during HF Progression

From the longitudinal experiments conducted, the normalised expression levels showed that MLCK gradually increased from 2D to 7D post-MI and decreased thereafter (Figure 6Aii,D), which corresponds with the compensatory and decompensatory changes in RLC phosphorylation profiles in post-MI samples. A significant increase was detected at 7D post-MI from 2D post-MI, where RLC phosphorylation was highest during HF progression. On the other hand, the expression levels of sham populations fluctuated and no clear pattern was observed (Figure 6Aii).

A gradual decrease in the normalised ZIPK expression levels was observed from 2D post-MI onwards in post-MI samples (Figure 6 Bii,E), suggesting that ZIPK does not directly contribute to the compensatory increase in RLC phosphorylation during HF progression. However, this may contribute to the decrease in RLC phosphorylation in late-stage HF. ZIPK expression levels of sham populations, on the other hand, seemed to increase very gradually from the timepoint of 2D post-MI, although the differences among the means of all timepoints were not statistically significant (according to ANOVA test).

The longitudinal normalised MYPT2 expression levels in post-MI samples depicts a profile that is inverse to normalised MLCK expression levels. It is interesting to note that the significant decrease in the expression levels at 7D post-MI directly relates to the largest compensatory increase in the RLC phosphorylation levels. On the other hand, the expression levels of sham population fluctuated, and no clear pattern was observed (Figure 6Cii,F).

To identify the key enzyme(s) responsible for the changes in RLC phosphorylation during HF progression, the changes in each enzyme expression level were compared to the changes in RLC phosphorylation at each timepoint. Focusing on post-MI samples, the expression levels of all three enzymes decreased at the 4D post-MI timepoint, corresponding to significant decreases in RLC phosphorylation levels from 2D post-MI timepoint (Table 2). The statistically significant decrease in MYPT2 was expected to cause an increase in the net phosphorylation. However, the reverse was observed. Thus, the decrease in the resultant phosphorylation implies that the coaction of MLCK and ZIPK has a greater effect than MYPT2. On the other hand, the expression levels of all three enzymes increased at the 28D post-MI timepoint, corresponding to a significant decrease in RLC phosphorylation. It is postulated that the statistically significant increase in MYPT2 caused a decrease in the net phosphorylation. In this case, however, the decrease in the resultant phosphorylation implies that the coaction of MLCK and ZIPK has a lesser effect than MYPT2, which contradicts the hypothesis made previously. Referring to another example in post-MI samples, at 14D and 28D post-MI, the significant increase and decrease in ZIPK expression levels, respectively, resulted in the phosphorylation levels decreasing and remaining unchanged, respectively. Thus, the significant changes in enzymes did not always bring about the expected phosphorylation effect.

Thus, the primary enzymes that brought about net RLC phosphorylation were unable to be elucidated based on the significant change in abundance, although the changes in the MLCK and MYPT2 expression levels were closely related to the RLC phosphorylation profiles. Nonetheless, these results clearly show that the net RLC phosphorylation is a complex interplay of various enzymes’ actions. Moving forward, it will be essential to explore the enzyme activity to elucidate the key enzyme(s) involved. A better understanding of the fundamental roles of enzymes such as ZIPK and MYPT2 will also allow for a more complete interpretation of these results. To isolate the roles of the individual enzymes, other forms of study models (e.g., causative models such as gene knockout) must be employed.

### 2.7. Differential RLC Expression in Left Ventricular Tissue during HF Progression

In this HF longitudinal study, the RLC amounts gradually increased from 2D to 7D post-MI and decreased thereafter (Figure 7B), while RLC amounts of sham samples were relatively stable (Figure 7C). There was a significant increase in RLC amounts at 4D and 7D post-MI from 2D post-MI (Figure 7B), which is defined as the compensatory phase in Section 2.3. Additionally, when the normalised amounts are superimposed (Figure 7D), the increase in RLC amounts at 7D post-MI is statistically significant. This opens another perspective of reasoning regarding the cardiac functional changes during HF progression (especially during the compensatory phase); that is, whether it is primarily due to the changes in RLC phosphorylation levels or the absolute amount of RLCs in the heart.

## 3. Discussion

Heart failure (HF) aetiologies include ischaemic (coronary artery disease, myocardial infarction), hypertensive (HFpEF—heart failure with preserved ejection fraction; HFrEF—heart failure with reduced ejection fraction) and primary cardiomyopathies (genetic, stress induced, etc.) [32,33]. However, epidemiological studies have shown that ischaemic heart disease (IHD) is a common cause of HF with a poor prognosis [34,35,36]. Thus, pathophysiological changes during ischaemia-induced HF progression were explored in this study with the use of the LAD-ligated MI mouse model. Past studies reported that hypertrophy was evident from 4D post-MI, that the ventricle wall became thinner from 7D post-MI [37] and that dilated hypertrophy develops from 4 weeks post-MI if the infarction is extensive [12] upon LAD ligation.

The choice of animal model and strain is extremely important to successfully model diseases, and in this case to recapitulate HF progression within a short time period. This is because species or different strains of a species can result in varying outcomes [38,39]. For example, even though decreased RLC phosphorylation levels are mostly associated with severe heart failure [40,41], there are studies that report that RLC phosphorylation continued to increase in end-stage HF in humans and rats [11,12]. On the other hand, when the LAD coronary artery was ligated to induce ischaemic injury, BalbC strain led to extensive thinning of the ventricular and septum walls, while 129S6 strain was most prone to infarct rupture by the endpoint of 28D post-MI. These results show that the different substrains can be used for studying different disease models [38]. C57BL/6 is useful for studying HF progression, as it displays characteristics (wall thinning, cardiac dilation, reduced ejection fraction) of a failing heart by 28D post-MI [38,39], while rats take longer at 20–32 weeks post-MI [42], where MI is induced by coronary artery ligation. C57BL/6 also displays relatively low incidence (36%) of myocardial rupture [38], making it useful for studying the functional and molecular changes over longer periods post-MI, with low attrition rates. Apart from the ability to manifest the characteristics of a failing heart post-MI, as presented in Section 2.1, the use of mice is advantageous due to their relative low cost, reproducibility, well-established genome and ability to be genetically modified [43]. LAD occlusion lower down in the coronary artery tree, referred to as LAD (L) in the present study, has an almost 0% fatality rate but only showed significant damage 8 weeks post-ligation. If the ligation position is too high, the fatality rate can be nearly 70%, although induction of visible MI damage is accelerated. In this study, LAD occlusion was performed at a high position, referred to as LAD (H), with ligation just above the bifurcation (1–2 mm below the tip of the left auricle). The results of surgery were reasonably repeatable and consistent, despite the fact that there were visible variations in the arrangement of the coronary tree from animal to animal.

Based on the hypertrophic index, hypertrophy was noticeable throughout HF progression. However, the gradual decrease in hypertrophic indices in the later stages indicated the possible dominance of cardiac dilation characteristics. The BNP expression levels were aligned with the signs of progressively failing human hearts, where the BNP levels were higher in hypertrophic cardiomyopathy as compared to dilated cardiomyopathy, with very low levels in control groups [29]. These features were aligned with features of failing human hearts, where hypertrophy was dominant in the initial stages as a compensatory mechanism to preserve cardiac function and dilation of the chambers in the later stages [44]. These were supported by the dilated phenotype observed from structural observation via histology.

Here, we provide a detailed time-course of changes in functional (Ca^2+^ sensitivity and cardiac contractility) and molecular (sarcomeric protein phosphorylation and enzyme expression quantification) characteristics, which have not been reported before. Dissecting the sequence of molecular changes during the time course is critical for treatment and drug development. For example, early medical interventions (within 24 h) include the use of angiotensin-converting enzyme inhibitors and the use of aldosterone antagonists for HF therapy following MI (within 7 days). However, early use (<24 h) of beta-blockers is associated with an increased risk of cardiogenic shock and death. Delayed beta-blocker administration post-MI is associated with a reduced risk of re-infarction and death [45].

Converging on Ca^2+^ sensitivity, it is the characterisation of the relationship between the calcium ions available for Troponin C binding and the amount of force generated by the muscle [30]. In the failing myocardium, calcium sensitivity has been reported to an either increase or decrease depending on the etiology of the disease [46]. In our study, calcium sensitivity and co-operativity increased (indicated by a lower EC_50_ and a higher Hill slope) at 2D and 4D post-MI groups when compared to its sham group. The gradient of the Hill slope indicates the Ca^2+^ dose response. Thus, a higher gradient indicates stronger co-operativity. However, there were no statistically significant differences in the data from 7D post-MI onwards, indicating a loose correlation between calcium sensitivity and contractile force, likely dependent on the stage of HF. In a previous study published by our lab, although the calcium sensitivity was not established, we found that the contractile force in the compensatory phase of HF (MI rat model) increased even in saturating calcium conditions (32 µM). This suggests that the increase in force compared tothe sham group could not be attributed to a change in calcium sensitivity [12]. On the contrary, other reports showed that mutations associated with hypertrophic cardiomyopathy (HCM) and restrictive cardiomyopathy (RCM) displayed increased Ca^2+^ sensitivity of force production, while mutations associated with dilated cardiomyopathy (DCM) demonstrated a decreased Ca^2+^ sensitivity of force production. One possible reason for myofibrillar Ca^2+^ sensitivity being higher in a failing heart than in a non-failing heart (with a 2–3-fold lower EC_50_) could be lower levels of Troponin I phosphorylation [47]. Another argument is that heart failure is associated with an early increase in Ca^2+^ sensitivity that reverts to decreased Ca^2+^ sensitivity at the end-stage of heart failure. Although we showed a pattern of increased and decreased Ca^2+^ sensitivity in our MI mouse model, these data are not conclusive and require further study [48]. However, we limit our interpretation to RLC phosphorylation to explain the change in contractile force at saturating calcium concentration.

To benefit from the advantages of RLC and its phosphorylated form, its role in HF development needs to be understood comprehensively. RLC phosphorylation levels are significantly reduced in heart failure patients [49,50,51], and the levels do not differ between dilated and ischaemic hearts [50,51]. On the other hand, in rats, the phosphorylation levels continue to increase at 20 weeks post-MI (decompensatory phase) [12], while rabbits experience a continuous decrease at 2 weeks post-MI [52]. Thus, there is no clear pattern in the variation of RLC phosphorylation during HF progression. Nonetheless, due to the ability of RLC phosphorylation to enhance contractility, it is postulated that RLC phosphorylation increases during the compensatory state and decreases during the decompensatory state. We also do not know whether RLC phosphorylation drives the functional changes during HF progression or whether the development of HF drives the RLC phosphorylation levels. Thus, the maximal isometric force potentiated by papillary fibres was measured at the different timepoints during HF progression. Papillary muscles closely represent the functionality of the heart, as studies showed that they lose contractility with MI and also thicken when the heart experiences hypertrophy [53]. Additionally, RLC phosphorylation in the left ventricular muscles was quantified. The longitudinal profiles of maximal isometric force and RLC phosphorylation levels during HF progression were superimposed. However, no significant time lag was identified, suggesting that RLC phosphorylation levels change transiently to enhance cardiac functionality during the compensatory phase of HF. This is contrary to the event of RLC phosphorylation in beating hearts where the levels are stable [54], suggesting a physiological change could have occurred due to MI.

The baseline RLC phosphorylation levels are regulated by the activities of cardiac myosin light chain kinase (MLCK) and myosin light chain phosphatase (MLCP). It is likely that alterations or disruption to the activation of these (or other) enzymatic pathways could have resulted in the deviation from the baseline levels. Supporting this, a study has reported that MLCK levels were significantly decreased 3 weeks post-MI in an HF mouse model [55]. The phosphorylation of cardiac isoforms occurs on a minute timescale, while smooth and fast skeletal RLCs are phosphorylated on a timescale of seconds in response to stimuli [54,56,57,58]. The heart is made of cardiac-specific fibres, with no other muscle isotypes, such as smooth, fast-twitch or slow-twitch skeletal muscle [59]. Other isoforms of RLCs are also not present in cardiac muscles [56]. This implies that the ability to capture the variations in phosphorylation profiles on a timescale of hours could be due to slower upstream signalling pathways. Even though the underlying mechanisms for these changes are unknown, it is apparent that the natural cardiac physiology has attempted to induce a compensatory effect to sustain cardiac function. The structural and contractile characteristics collectively indicate the compensatory phase to be between 4D and 7D post-MI, while the decompensatory phase is between 10D and 28D post-MI.

Our results also show that when phosphorylation levels were between 0.35 and 0.42 mol Pi/mol RLC, the isometric force was sustained at physiologically appropriate levels. This range is close to the baseline RLC phosphorylation levels of ~0.3–0.4 mol Pi/mol RLC, suggesting that maintaining RLC phosphorylation levels to baseline levels could recover cardiac contractility. It has been established that enhanced RLC phosphorylation levels in permeabilized striated muscle fibres improve muscle contractility and phosphorylation of R58Q, while A13T mutated RLC restores Ca^2+^ binding to RLC [57]. Similarly, when the procedure was employed in 14D post-MI permeabilized papillary fibres, the phosphorylation levels increased by 72.6% and the force of contraction increased by 57%. This finding was key in substantiating the possibility of using phosphorylated RLCs in cardiac disease treatment by improving cardiac contractility, as contractility alongside heart rate substantiates the basic functionality of the heart [58].

This study also opens up another perspective of reasoning regarding the cardiac functionality changes during HF progression relating to the changes in the amount of RLCs during HF progression. A study published in 1992 showed that RLC amounts in human hearts were significantly reduced in idiopathic dilated cardiomyopathy, possibly due to protease-mediated cleavage of RLC [60]. Another study showed that the ELC/RLC ratio was maintained between healthy and failing hearts. However, there were no comments on the absolute amounts of RLCs [50]. In our results, the RLC amounts were enhanced in the compensatory phase, suggesting that the increase in RLC amounts could improve cardiac function. It is known that RLC levels are precisely regulated by post-transcriptional mechanisms based on a study of overexpressed RLC mRNA and unchanged RLC protein levels [61,62]. The reasons underlying the changes in RLC expression during HF progression are unclear. However, based on the above-mentioned studies, the changes could be due to a disruption in the protease-mediated cleavage or post-transcriptional cellular mechanisms.

The downstream enzymes (kinases and phosphatases) play a key role in regulating phosphorylation levels. On that note, ZIPK, MLCK and MLCP are known to be directly involved in regulating RLC phosphorylation. According to the literature, the levels of MLCK and ZIPK are expected to decrease during HF progression, while the patterns of MLCP expression during cardiac disease or HF have not been reported [9,55,63]. It is established that basal cardiac RLC phosphorylation levels are regulated primarily by cardiac myosin light chain kinase (MLCK) and myosin light chain phosphatase (MLCP) in humans and animals such as mice, rats and pigs [15,64,65]. In MLCK knockout mice, the cardiac RLC phosphorylation levels decreased to 10% and heart failure was developed [15]. In MI mouse models, both cardiac MLCK and RLC phosphorylation levels were reduced in HF tissues (3 weeks post-MI) as compared to control [55]. While MLCK mRNA levels in mice were sustained in failing hearts [55], the levels increased in humans and rats [65], indicating variations in the transcription process. Our results show that MLCK expression is generally increased until 7D post-MI and decreases thereafter, which corresponds to RLC phosphorylation profiles. However, the activity of these enzymes needs to be monitored to elucidate the underlying pathways.

Studies have shown that other kinases are also able to phosphorylate RLCs. A prominent enzyme is ZIPK, a 52.5 kDa serine/threonine kinase, also known as death-associated protein kinase 3 [66], which is ubiquitously expressed in human and mouse hearts [67,68]. Ventricular RLC is a prominent substrate of ZIPK, as identified by an unbiased substrate search on heart homogenates [69]. Biochemical analysis confirmed that ZIPK phosphorylates the Ser-15 residue in mouse and human RLCs [69]. ZIPK also regulates RLC phosphorylation levels by activating myosin light chain phosphatases via direct phosphorylation [70]. However, its activation and role in cardiomyocytes are unknown [69]. The changes in basal RLC phosphorylation levels were not associated with any changes in ZIPK levels [71]. However, lower levels of ZIPK expression were observed in the hearts of patients with dilated cardiomyopathy [9,63]. Aligned with the literature, our results demonstrate that ZIPK expression continually decreases during HF progression. Our results also show that ZIPK expression does not directly contribute to the increase in RLC phosphorylation during the compensatory stage.

The baseline RLC phosphorylation levels in healthy hearts are maintained as MLCPs work concurrently [15] with the kinases. Striated muscle MLCPs are composed of a regulatory targeting subunit (MYPT2, 110–130 kDa), a catalytic subunit (PP1cδ, 38 kDa) and a small subunit (hHS-M_21_) with an unknown function [54,72,73]. MYPT2 interacts with PP1cδ to target myosin filaments and potentiate phosphatase activity in striated muscles [74,75]. In wild-type mice, 100% of the MYPT2 was constitutively and maximally phosphorylated at Thr646 (responsible kinase is unidentified) in beating hearts [15,76,77]. This constitutive phosphorylation is predicted to be an intramolecular autoinhibition mechanism [78]. The MYPT2 dephosphorylation mechanism and the profile of phosphorylated MYPT2 in diseased hearts have not been elucidated yet [15,76]. In this study, when both wild-type and post-MI groups were resolved in phos-tag gels and blotted, only one band of protein was observed, meaning all MYPT2 types are assumed to be phosphorylated. In our results, MYPT2 expression gradually decreased until 7D post-MI, when RLC phosphorylation was the highest, and then increased thereafter.

The quantification and correlation of enzymes collectively suggest that MLCK and MYPT2 regulate RLC phosphorylation through the compensatory and decompensatory phases, while ZIPK potentially contributes to the decompensatory phase only.

Cardiac RLCs are also phosphorylated by protein kinase C (PKC) via the activation of the α-adrenergic pathway. PKC phosphorylates at different sites from MLCK [79]. CaM-dependent kinase II (CaMKII) phosphorylates RLCs under inotropic effect [80]. Rho-kinase (ROCK) indirectly (via stimulation of α-adrenergic receptors) increases RLC phosphorylation levels by inhibiting MLCP activity via phosphorylation of a fragment (MYPT2) [14,81]. However, these kinases do not only phosphorylate RLCs but also other sarcomeric proteins, such as troponin I (TnI) and cardiac-myosin-binding protein C (MyBP-C).

On that note, it is undeniable that other sarcomeric proteins co-regulate muscle contraction, and it is important to study their role in HF progression. Our results suggest that RLC phosphorylation is directly correlated while MyBP-C phosphorylation is inversely correlated to isometric force. The latter is contrary to the current understanding of MyBP-Cs. Most studies report a decrease in MyBP-C phosphorylation levels in end-stage HF. For example, mice that had been genetically or surgically (transaortic constriction/ischaemic-reperfusion) modified to develop stressed hearts and failing human myocardium displayed increased dephosphorylated MyBP-C levels, which were related to contractile dysfunction and heart failure [82,83,84,85]. However, on the other hand, another study with an MI rat model displayed increased cMyBP-C phosphorylation levels in the decompensatory phase at 4 weeks post-MI and 20 weeks post-MI, respectively [12]. Similarly, the MyBP-C phosphorylation levels increased from 14D post-MI onwards in this longitudinal study. The rationale behind these discrepancies is unclear. However, a possible reason could be differences in the strains of mice or species used for experimentation, resulting in differential G-protein coupled receptor activation [86,87]. Nonetheless, the decrease in MyBP-C phosphorylation (where phosphorylation is known to induce ionotropic effects) during the compensatory phase is unexplainable at the moment. On the other hand, the opposite phosphorylation profiles of RLC and MyBP-C during HF progression are thought-provoking, as they present the possibility of phosphorylation activity being affected by spatial hindrance due to the alignment of the molecules [88].

Our results show no particular change in TnI phosphorylation levels during HF progression. This is similar to a rat ischaemic mouse model presented in another study [12]. However, TnI phosphorylation levels are significantly decreased in failing human hearts [85,89,90]. TnI is phosphorylated by multiple kinases, and these kinases are activated by various signalling pathways [91]. Some of these kinases phosphorylate specific sites activating ionotropic effects, while other kinases phosphorylate other sites producing an opposite effect, such as a reduced cross-bridge cycling rate [91,92]. For example, phosphorylation of Ser22/23 of TnI by PKA in mice increases the cross-bridge cycling rate [91], while phosphorylation of Ser43/45 of TnI by PKC decreases the cross-bridge cycling rate [91]. Thus, there is a possibility that the almost stabilised phosphorylation levels between 4D and 28D post-MI represent an attempt of the kinases and phosphatases to achieve a phosphorylation profile to induce inotropic (compensatory) effects [91,92]. There is also evidence of desensitisation of G-coupled protein receptors during heart failure, which affects the activation of downstream enzymes [90]. These could have influenced the resultant phosphorylation profile.

The complexity of the underlying mechanisms resulting in the compensatory and decompensatory phases is extensive. Moreover, proteomic and phosphoproteomic studies on cardiac muscle during HF [93,94] have not been extensively explored. This is important to establish therapeutics for cardiac diseases. In a recent publication, a potential treatment by altering cross-bridge dynamics using Mavacamten (oral treatment targeting cardiac specific myosin) was reported. This was demonstrated on muscle fibres isolated from hearts experiencing hypertrophic cardiomyopathy [95,96]. Similarly, the in vitro effects of the phosphorylation of RLC or other sarcomeric proteins can be appropriately exploited to manipulate cardiac muscle function in vivo. However, more investigation is needed on the cellular mechanisms prior to progressing to advanced in vivo models.

On a technical front, our experience and those of others show that it is inaccurate to rely on an internal control protein for C57BL/6 mouse MI models [97]. This is because common housekeeping proteins such as GAPDH and β-actin express differentially at different stages of ischaemic myocardial injury [97]. Similarly to the literature, in our experiments, the GAPDH amounts are generally lower in post-MI groups as compared to their age-matched controls, with statistically significant differences between 14D post-MI and age-matched control populations (Appendix A). Hence, equal amounts of total proteins, as calculated by BCA protein assay kit, were loaded per lane as loading controls. Besides using total proteins as loading controls, phosphorylation results were obtained by normalising the phosphorylated protein amounts by their respective total protein amounts. For experiments studying the protein abundance, the protein intensities were normalised to a common factor (e.g., 2D sham or 2D post-MI) within the blot. These measures were taken to ensure reliable and accurate representation of the results.

## 4. Materials and Methods

### 4.1. Generation of Chronic Myocardial Infarction (MI) Mouse Model

Animal experiments were carried out in accordance with the guidelines of the University Institutional Animal Care and Use Committee (IACUC, ARF-SBS/NIE-A0354), Singapore. C57/BL6 male mice aged 4–5 weeks were used for data collection. In preparation for surgery, the mice were placed supine on a temperature-regulated surgical table at 37 °C ± 0.5 °C. Ventilation via endotracheal tube was maintained with 100% oxygen using a small animal ventilator (Hugo Sachs, March, Germany) delivering a tidal volume of 7 µL/g at 200 breaths/min. An incision in the skin was made left of the prominent xiphoid cartilage. Skin and muscle layers were gently separated to expose the 3rd and 4th ribs. The intercostal space was opened following the natural angle of the ribcage and retracted to expose the heart. The pericardium was gently picked up and pulled apart. The LAD coronary artery was then identified, and with a 9–0 suture, ligated with three knots approximately above the bifurcation (H), as shown in Figure 1A. The bottom left portion of the left ventricle clearly bleached and the heart rate increased for a short period (~3–5 s) upon successful ligation. The chest cavity was closed with one or two 5–0 nylon sutures (Greenwood R&D support Pte. Ltd., Singapore) and the muscles and skin were closed layer-by-layer with 5–0 nylon sutures. A 1 mL syringe attached to a 25-gauge needle was then inserted about 5 mm below the ribcage to remove the air trapped in the chest cavity by pulling the plunger gradually. This helped the mice to develop a regular breathing pattern and prevented the occurrence of pneumothorax. Buprenorphine (0.1 mg/kg) was intramuscularly injected into the mice at a dose of 0.1 mL/10 g to manage post-operative pain [98]. The intubation needle was removed when the effects of isoflurane subsided. This was confirmed by gently pinching toes to check for recovery of reflex responses, such as pedal withdrawal [99]. Buprenorphine was administered for the next two days to ease any pain experienced by the mice. The sham-operated mice underwent the same procedure except for tying knots to occlude the artery.

### 4.2. Papillary Preparation for Maximal Isometric Force and Ca^2+^ Sensitivity Measurements

Papillary muscles were used as surrogate cardiac muscle in the undamaged region of the heart. The anterior and posterior ventricular papillary muscles from mouse hearts were isolated according to the methods described before with minor modifications (see Appendix A) [12]. Briefly, each isolated papillary muscle was divided into 3 segments measuring 2 mm in length and 150 μm in diameter. The ends of the muscle segments were crimped with aluminium T-clips and were held at just beyond slack length. The segments were then permeabilised with 1% Triton overnight. This resulted in the perforation of the cellular and organelle membranes, allowing chemicals to penetrate. The demembranation also resulted in the loss of enzymes and enzyme-mediators. Thus, the phosphorylation status of the sarcomeric proteins was unaffected by the permeabilisation process [100,101].

Prior to force measurements, the permeabilised papillary segments were attached to the experimental rig by hooking the ends to a customised force transducer (AE801, Kronex, Oakland, CA, USA) and a servomotor in a 6-trough stage [11]. The initial force (normalised to the cross-sectional area of the muscle segment) of all segments in the relaxed state was offset to 2 kN/m^2^ in relaxing solution by adjusting the length of the preparation. In practice, the muscle segments were allowed to stabilise in relaxed buffer for 3 min before being transferred to the pre-active buffer. The muscle segments were transferred to the pre-activating buffer by rotating the stage so that the length remained and was incubated for 3 min to prime the muscles for active contraction. For the measurement of maximal isometric force, the muscle segments were transferred to an experimental trough containing activating solution with 5 mM Mg^2+^ ATP and saturating calcium concentration (32 μM) and then incubated for 3 min (to ensure the active buffer is optimally diffused into the muscle fibre). By then, the plateau of the maximal isometric force was reached and recorded using LabVIEW software (National Instruments Corp, Singapore). The average force values from three contractions were calculated for each papillary muscle segment. The detailed buffer formulations are provided in the Appendix A. Additionally, the method used to characterise Ca^2+^ sensitivity is also described in the Appendix A.

### 4.3. Measurement of Sarcomeric Protein Phosphorylation

During cardiac dissection for papillary muscle isolation, the left ventricles were excised and snap frozen in liquid nitrogen prior to storing them at −80 °C. The frozen ventricles were mechanically pulverised using a mortar and pestle and added to Laemmeli sample buffer (62.5 mM tris, 2% SDS, 10% (*v*/*v*) glycerol, 50 mM dithiothreitol (DTT), 6.7 M Urea, pH 6.8) at a ratio of 20 µL/mg. The lysate was mixed and quickly heated at 100 °C for 5 min to denature proteases and phosphoproteases. Prior to conducting gel electrophoresis, the concentration of the lysate proteins was determined using a Micro BCA Protein Assay Kit (Thermofisher Scientific, Waltham, MA, USA). Phos-tag polyacrylamide (Nard Institute Ltd., Japan) gel was used to quantify sarcomeric protein phosphorylation levels. The resolving and stacking gels (Appendix A) were prepared and left to set for 90 min and 45 min, respectively. Each well was loaded with 300 µg of lysate proteins. Gel electrophoresis was conducted at 140 V for 150 min at room temperature. The electrophoresis tanks were wrapped with cold (~4–15 °C) wet cloth to facilitate heat transfer away from the buffer, as phos-tag gels are sensitive to high temperature and cause distortion to resolving protein bands as a result [102]. Prior to protein transfer, the phos-tag gels were treated with transfer buffer (25 mM tris, 192 mM glycine, 20% (*v*/*v*) methanol, pH 8.3) containing 10 mM EDTA for a total of 20 min (two sets of 10 min treatment with a change of buffer in between) to improve transfer efficiency. After this, the gel was equilibrated in EDTA free-transfer buffer for a total of 20 min (two sets of 10 min treatment and change of buffer in between). A pore-sized PVDF membrane measuring 0.2 µm was used for transferred proteins to adhere to. PVDF membranes were soaked in 100% methanol for activation. Proteins were transferred at 400 mA for an hour at 4 °C. Blots containing transferred proteins were blocked in 5% BSA in TBST for an hour and then incubated with 1° antibodies according to the recommended dilutions in the product specifications. The incubation was carried out overnight at 4 °C. The next day, excess antibodies were removed by washing the blots with TBST for 30 min (change of buffer every 10 min). The blot was then incubated in 2° antibodies conjugated with HRP, for an hour at room temperature. After this, the blots were exposed to chemiluminescence reagents (Bio-Rad Laboratories, Singapore). The oxidation of the reagents was catalysed by HRP, resulting in a chemiluminescence signal (light at 425 nm) being emitted [103]. This signal was then detected by the BioRad ChemiDoc MP-Imaging system and an image was captured immediately using the ImageLab5.2 software (Biorad, Singapore). The formula to calculate the phosphorylation is stated below. The details of the antibodies used are presented in Appendix A. A comprehensive guide for using Phos-tag gels is provided in [104].
$$\mathrm{Ratio}\;\mathrm{of}\;\mathrm{RLC}\;\mathrm{phosphorylation}=\frac{intensityof\left(1X1P\right)+intensityof\left(2x2P\right)}{intensityof\left(0P+1P+2P\right)}$$

### 4.4. Quantifications of Enzymes That Regulate RLC Phosphorylation

The methodology used to quantify enzyme expression was exactly the same as described previously in Section 2.3. However, an ordinary polyacrylamide gel was used instead of phos-tag gel. A proportion of 15% polyacrylamide gel was used to resolve MLCK and ZIPK, while 10% was used to resolve MYPT2. The constituents of the gels are presented in Appendix A.

### 4.5. Expression and Purification of Recombinant RLC (rRLC)

The rRLC plasmid was donated by Professor Irving’s laboratory at King’s College London. The expression and purification protocols were described in a previous study and are detailed in the Appendix A [11].

### 4.6. In Vitro rRLC Phosphorylation

Concentrated RLCs (1 mg/mL) were mixed with phosphorylation kinase buffer (PKB) (25 mM HEPES, pH 7.1, 1 mM MgCl_2_, 200 mM NaCl, 1 mM DTT, 0.2 mM Na_2_ATP, 5 mM glycerol), 10 µg myosin light chain kinase (MLCK, Sigma M9197) and 10 µg zipper-interacting protein kinase (ZIPK, Sigma D7194, St. Louis, MO, USA) at a volume ratio of 800:200:1:8 (RLC:PKB:MLCK:ZIPK). The mixture was then incubated at 37 °C for two hours, with MLCK and ZIPK being replenished with the same volume every half hour. The phosphorylated RLCs were resolved in 10% polyacrylamide gel containing 25 µM phos-tag molecules for 45 min at 180 V under room temperature, followed by the above-mentioned Western blotting procedure. RLC phosphorylation levels were calculated using ImageJ, showing that 51.7% of the recombinant RLCs were phosphorylated (figure not shown). After this, they were aliquoted into small fractions and stored at −80 °C for long-term use.

### 4.7. Phosphorylated rRLC Exchange

Exchange buffer (10 mM Na_2_ATP, 20 mM Imidazole, 10 mM EGTA, 10 mM EDTA, 20 mM KH_2_PO_4_, 300 mM potassium propionate, 0.5 mM trifluoperazine, 5 mM DTT, pH 6.5), phosphorylated rRLC and TnC (LeeBio, Maryland Heights, MO, USA) were required for the exchange procedure. The use of the above-mentioned chemicals, primarily involving trifluoperazine, induced a conformational change in native RLCs, resulting in it dissociating from the myosin heavy chains [105], allowing phosphorylated recombinant RLCs to occupy the site. TnC is significantly reduced during permeabilisation and was replenished to preserve muscle contractility [11,14,106]. The exchange procedure was conducted at 20 °C. Permeabilised papillary muscle segments were incubated for a total of 45 min in exchange buffer containing 800 µM of phosphorylated rRLC and 57.8 µM of TnC with a replenishment of 30 µL of exchange buffer (containing TnC and phosphorylated RLC) every 15 min. To remove unbound phosphorylated rRLC after the exchange procedure, the fibres were incubated in 57.8 µM TnC containing relaxing buffer for 10 min at 20 °C. In the existing literature, the amounts of RLC used for exchange range between 30 µM and 25 mM [11,107,108]. However, the exchange efficiency is comparable when 30 µM, 800 µM or 25 mM of RLC is used [11]. To calculate the exchange efficiency, five fibres underwent exchange with 1° Rho·RLC only and another five underwent 1° Rho·RLC and 2° unlbl·RLC exchange (Appendix A). The proteins from the fibres were extracted with Laemmli buffer (3 µL/fibre) and resolved with 10% polyacrylamide–SDS gel for 1 h at 140 V (room temperature). The signal from the HRP-conjugated RLC was captured with the BioRad ChemiDoc MP-Imaging system. The exchange efficiency was calculated using the formula below. Using the above-mentioned protocol, the exchange efficiency of RLCs in permeabilised papillary segments was 47%, as presented in Appendix A.
(1)$$\mathrm{Exchange}\;\mathrm{efficiency}=\frac{1{}^{\circ}Rho\cdot RLCsignal-\left(1{}^{\circ}Rho\cdot RLC\wedge 2{}^{\circ}unlbl\cdot RLC\right)signal}{1{}^{\circ}Rho\cdot RLCsignal}$$

### 4.8. Statistical Analysis

Statistics were calculated using GraphPad Prism 7.2 (GraphPad Software, San Diego, CA, USA). The normality of the sample populations was determined using Shapiro–Wilk’s test and homogeneity was determined using the F-test, prior to conducting Student’s *t*-test. One- or two-tailed Student’s *t*-tests were performed to determine statistical significance between the means of the two groups. The normality of the sample populations was determined using Shapiro–Wilk’s test and homogeneity by F-test, prior to conducting Student’s *t*-test. If the sample populations were nonparametric, Mann–Whitney test was employed; if they were parametric but not homogenous, Student’s t-test with Welch’s correction was applied. Here, * *p* < 0.05, ** *p* < 0.01, *** *p* < 0.001

Ordinary one-way ANOVA was used to determine statistical differences among the means of more than two sample populations that were parametric and homogenous. Brown–Forsythe or Bartlett’s test were performed to verify homogeneity of results prior to ANOVA test. If the sample populations were nonparametric, Kruskal-Wallis test was employed. If the samples were parametric but not homogenous, Brown–Forsythe or Welch’s ANOVA test was employed. All data are shown as means ± S.E.

## 5. Conclusions

This study suggests that RLC phosphorylation changes transiently to bring about a compensatory effect of cardiac function. RLC phosphorylation also has the potential to reverse contractile impairment stemming from cardiac disease. There are likely other key players altering muscle contraction during HF progression. Thus, the interactions between RLCs and the other players need to be elucidated to design an effective treatment for HF.

## Figures and Tables

**Figure 1 ijms-23-00088-f001:**
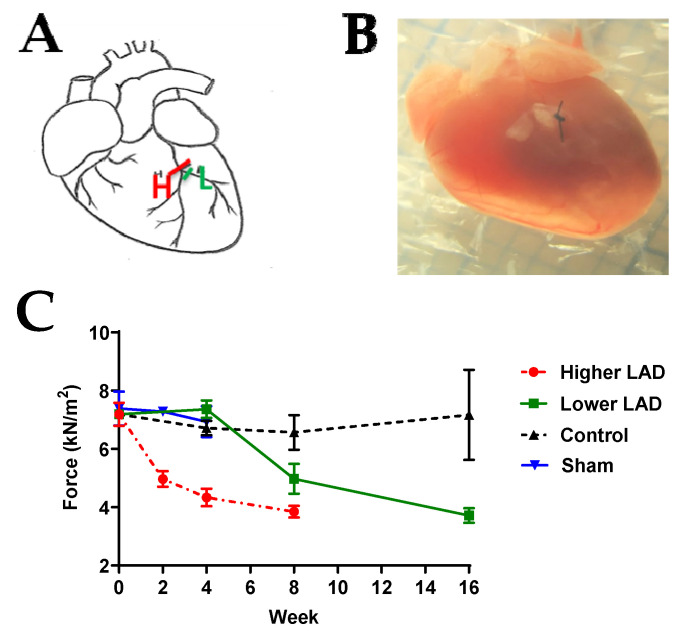
LAD ligation optimisation. (**A**) Schematic diagram of the ligation location. H: ligation just above the bifurcation, L: ligation at the left branch just below the bifurcation. (**B**) Image of the mouse heart with ligation at H. (**C**) Longitudinal comparison of contractile force following LAD (H), LAD (L) surgery, control (no surgery) and sham surgery groups. Note: *n* = 3–13, where ‘*n*’ refers to the number of muscle preparations in each group from a total of 2–6 mice. Statistically insignificant differences among the means, verified by ordinary one-way ANOVA test.

**Figure 2 ijms-23-00088-f002:**
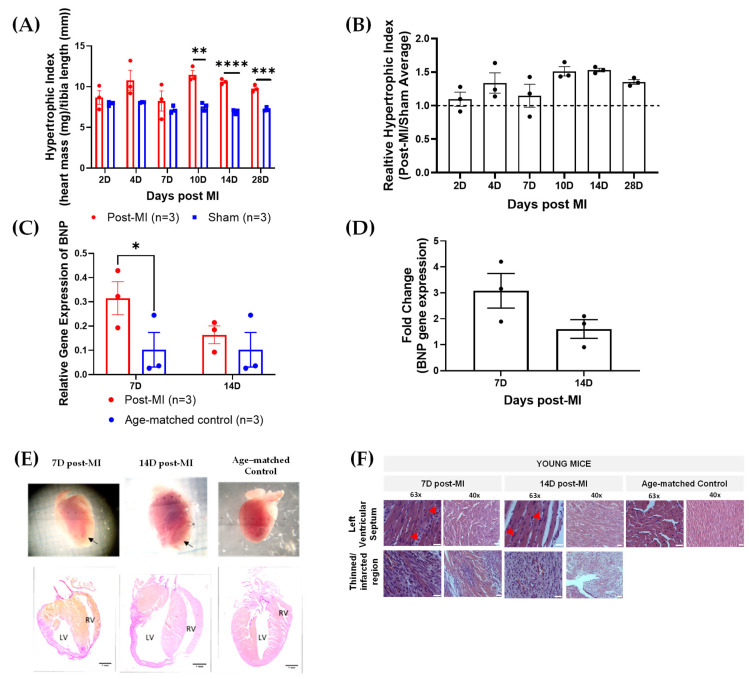
Physiological, biological and structural markers verifying HF progression. (**A**) Raw hypertrophic indices of post-MI and sham groups between 2D and 28D post-MI. (**B**) Relative hypertrophic indices (post-MI/sham average) of post-MI groups between 2D and 28D post-MI. (**C**) Relative BNP gene expression levels at 7D and 14D post-MI and sham groups. (**D**) Fold changes in gene expression of post-MI populations with respect to their sham groups. Note: ‘*n*’ refers to the number of mice. Unpaired one-tailed *t*-test used to determine the differences between post-MI and sham groups. Note: * *p* < 0.05, ** *p* < 0.01, *** *p* < 0.001, **** *p* < 0.0001. (**B**) No significant differences among the means were verified using ordinary one-way ANOVA. (**E**) Heart images and histology (Van Gieson stain) of longitudinal heart sections show that the heart displays features of hypertrophic cardiomyopathy at both 7D and 14D post-MI and dilated cardiomyopathy at 14D post-MI. Black arrows point to the site of thinned left ventricular walls. Images are taken at 10× magnification. Scale bar = 1 mm. Acronym: RV, right ventricle; LV, left ventricle; VS, ventricular septum. (**F**) H&E-stained, longitudinal heart sections at 7D and 14D post-MI and age-matched control, indicating post-MI features. Red arrows indicate site of inflammatory cell infiltration. Images were taken at 63× oil immersion and 40× magnification. Scale bar = 25 µm.

**Figure 3 ijms-23-00088-f003:**
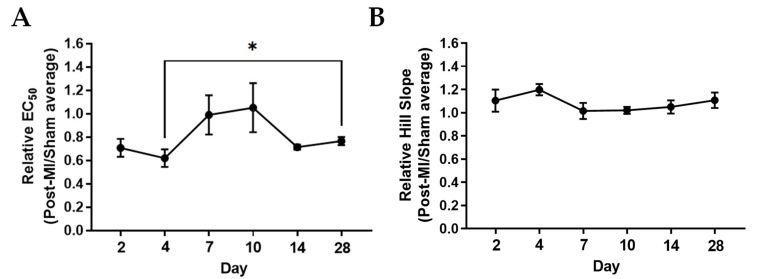
Ca^2+^ sensitivity and cooperativity changes during MI progression. (**A**) Relative EC_50_ and (**B**) relative Hill slopes (post-MI/sham average) plotted by mouse. Each data point is the average of at least three muscle preparations from the cardiac papillary of one mouse. Note: *n* = 3–5, where ‘*n*’ refers to the number of mice used. One-tailed unpaired t-test was performed between timepoints. Note: ** p* < 0.05 Kruskal–Wallis test was used on EC_50_ of samples and ordinary one-way ANOVA was used on Hill slope of samples. There were no significant differences among the samples of the respective groups.

**Figure 5 ijms-23-00088-f005:**
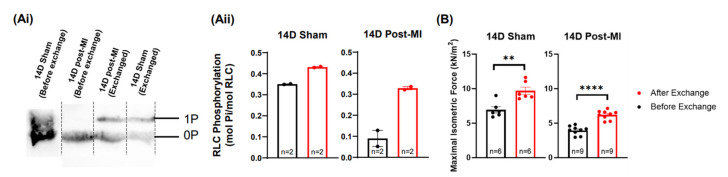
Potential for RLC phosphorylation in terms of muscle recovery. (**Ai**) Blot images of RLC extracted papillary muscle fibres under two treatments: 14D sham and post-MI permeabilized fibres (control) and 14D post-MI and sham permeabilized fibres exchanged with phosphorylated RLCs. Each lane contains protein samples extracted from five papillary fibre segments. (**Aii**) Bar graph representation of phosphorylation levels for the above-mentioned groups. Unpaired one-tailed Mann–Whitney test was performed for the respective groups, but no significant difference was detected between the ranks. Note: ‘*n*’ refers to the technical replicates. Two mice were used for post-MI groups and one mouse was used for sham group. (**B**) Bar graph of maximal isometric force of the above-mentioned groups. Unpaired one-tailed *t*-test was conducted to test for statistically significant differences. Three mice were used for each group. Note: ‘*n*’ refers to the number of papillary segments used for data collection; ** *p* < 0.01, **** *p* < 0.0001.

**Figure 6 ijms-23-00088-f006:**
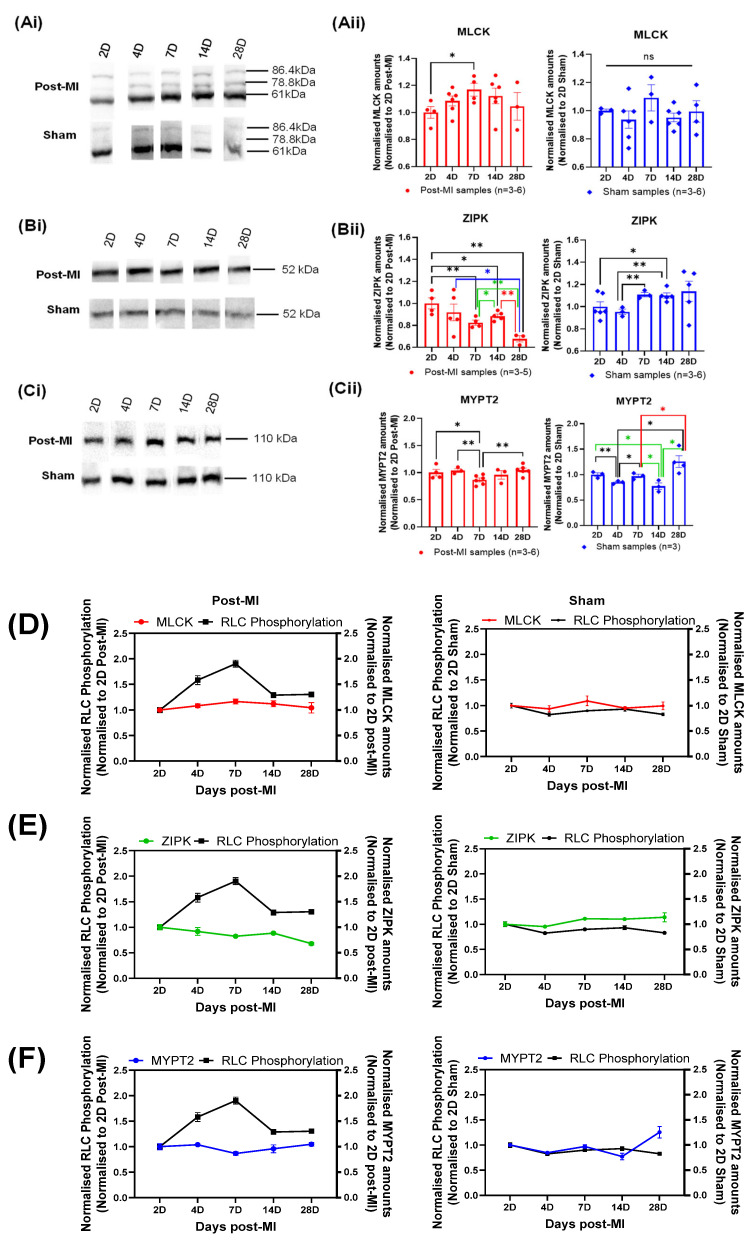
Longitudinal changes in enzymes in mice. (**Ai**,**Bi**,**Ci**) The blot images of MLCK, ZIPK and MYPT2, respectively. (**Aii**,**Bii**,**Cii**) Bar graph of normalised MLCK, ZIPK and MYPT2 amounts of post-MI and sham populations, respectively. All sample populations are normally distributed and validated by Shapiro–Wilk test. Differences in the standard deviations between two timepoints were tested by F-test. Unpaired one-tailed *t*-test with Welch’s correction (*) was performed on those with significant differences in the variances, while unpaired one-tailed *t*-test (*) was performed for all other samples. Note: *n* = 3–6, derived from at least two mice for post-MI and three mice for sham (except 7D sham, where one mouse was used). Note: ‘*n*’ refers to the number of technical replicates; * *p* < 0.05, ** *p* < 0.01. (**D**–**F**) Line graph plotted with normalised enzyme values and RLC phosphorylation to depict longitudinal changes in post-MI and sham populations.

**Figure 7 ijms-23-00088-f007:**
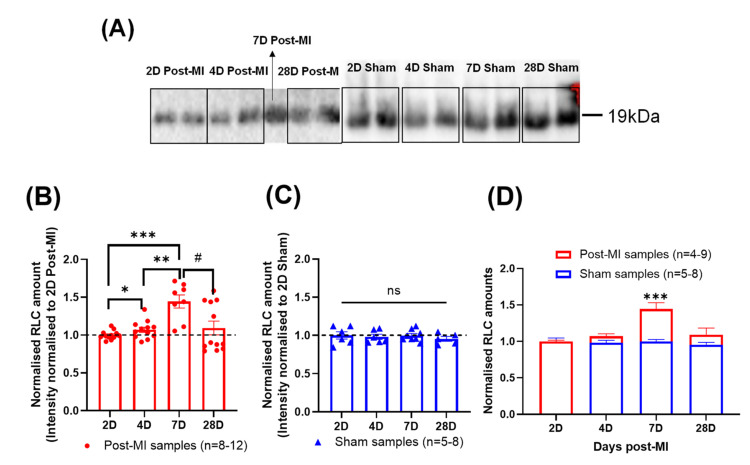
Changes in the quantity of RLCs suggesting losses in diseased mouse sarcomere. (**A**) Blot image of RLC of post-MI and sham groups at different time points ranging between 2D and 28D post-MI. (**B**) Bar graph of normalised RLC amounts at different time points for post-MI groups. Unpaired one-tailed t-test with Welch’s correction (*) was used between 2D and 4D, 2D and 7D and 4D and 7D samples. Mann-Whitney one-tailed test (#) was performed between 7D and 28D samples. Kruskal-Wallis test indicated that the medians varied significantly among samples (*p* = 0.0194). All samples were normally distributed (except 28D post-MI), as validated by Shapiro-Wilk test. There were significant differences in the standard deviations, except between 4D and 7D post-MI and 7D and 28D post-MI, as validated by F-test. Results were obtained from at least two mice for post-MI samples. Note: *n* = 4-9, where ‘*n*’ refers to the number of technical replicates; * *p* < 0.05, ** *p* < 0.01, *** *p* < 0.001, ^#^
*p* < 0.05. (**C**) Bar graph of normalised RLC amounts at different time points for sham groups. Ordinary one-way ANOVA showed no significant differences among population means. Sample populations were normally distributed, as validated by Shapiro-Wilk test. No statistical significance detected between timepoints, validated by F-test. Results were obtained from one mouse for sham samples. Note: *n* = 5-8, where ‘*n*’ refers to the number of technical replicates. (**D**) Superimposed graph of post-MI and sham samples. Unpaired one-tailed *t*-test with Welch’s correction was performed on 7D samples.

**Table 1 ijms-23-00088-t001:** EC_50_ and Hill slopes of post-MI and sham groups.

Post-MI Group							Sham Group
Time-Point	2D	4D	7D	10D	14D	28D	Average of All Time Points
${\mathrm{EC}}_{50}$ (µM)	1.02 ± 0.11	0.898 ± 0.11	1.43 ± 0.24	1.51 ± 0.3	1.03 ± 0.03	1.10 ± 0.05	1.34 ± 0.09
Hill Slope	1.48 ± 0.26	1.50 ± 0.11	1.2 ± 0.18	1.27 ± 0.19	1.28 ± 0.08	1.44 ± 0.19	1.25 ± 0.19

**Table 2 ijms-23-00088-t002:** Summary of changes in the enzyme expression levels in post-MI and sham populations of mice. Arrows represent increases or decreases in normalised enzyme expression or RLC phosphorylation with respect to the previous timepoint. Note: ‘-’ indicates no change from previous timepoint. Asterisks indicate significant changes from the previous timepoints, validated by unpaired one-tailed *t*-test or with Welch’s correction. Note: * *p* < 0.05, ** *p* < 0.01, **** *p* < 0.0001.

Sham Samples	4D	7D	14D	28D
MLCK	↓	↑	↓	↑
ZIPK	↓	↑(**)	-	↑
MYPT2	↓(**)	↑(*)	↓(*)	↑(*)
RLC Phosphorylation	↓(**)	↑(*)	↑	↓(*)
**Post-MI samples**	**4D**	**7D**	**14D**	**28D**
MLCK	↑	↑	↓	↓
ZIPK	↓	↓	↑(*)	↓(**)
MYPT2	↑	↓(**)	↑	↑
RLC Phosphorylation	↑(****)	↑(*)	↓(****)	-

## Data Availability

The data presented in this study are available on request from the corresponding author.

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
