# Peer review of "Functional and Molecular Characterisation of Heart Failure Progression in Mice and the Role of Myosin Regulatory Light Chains in the Recovery of Cardiac Muscle Function"

_ijms, 2021, doi:10.3390/ijms23010088_

Round 1

Reviewer 1 Report

I have no further comments to the authors.

Author Response

Dear Reviewer, 

Many thanks for taking time to review our paper. 

Regards,

Kasturi

Reviewer 2 Report

Review comments attached

Author Response

Dear Reviewer,

Many thanks for taking time to review our manuscript.

Regards,

Kasturi

Round 2

Reviewer 2 Report

Hi,

Thank you for responding to my concerns.

My major concern now is you protein normalization and blots. Well, I agree that you have normalized the protein concentration by quantifying and loading equal amounts but when you are comparing sham vs. MI samples, the samples must be loaded in the same gel to avoid errors of unequal loading and transfer. If there are large number of samples then you could have split the samples by running 2D, 7D in one gel (sham and MI) and the remaining in another gel. As your results and interpretation mostly rely on protein phosphorylation, it becomes important to keep these factors into consideration.

Also, as you have mentioned in the response letter that some of these sample have been run together, so please replace these blots with the original blot for better representation rather than showing single lanes.  Representing single lane is not a good representation so please provide entire blot.

Author Response

Dear reviewer,

Many thanks for your comments. I have addressed it below. 

Regards,

Kasturi

Round 3

Reviewer 2 Report

Thank You for the reply

This manuscript is a resubmission of an earlier submission. The following is a list of the peer review reports and author responses from that submission.

Round 1

Reviewer 1 Report

Comments

This is a longitudinal study on sarcomeric protein changes in mediating compensatory and decompensatory transitions following MI in mice. In brief, they have studied the changes of maximal isometric force, calcium sensitivity (EC50) and regulatory light chain (RLC) phosphorylation between 2 days to 28 days post-MI. They have incorporated native RLC with recombinant phosphorylated RLC in skinned diseased papillary fibres and measured the corresponding muscle contraction. In addition, the authors have examined the expressions of MLCK, ZIPK, MYPT2, those regulate cardiac RLC phosphorylation as well as other regulatory sarcomeric proteins including TnI and MyBP-C. The study has identified RLC and MYBPC phosphorylations and regulatory enzymes in mediating contractile functions, suggesting therapeutic targets of preventing HF.

I have a few comments which may help to improve the study:

  • The authors suggested LAD (H) as the protocol for MI study. It would be useful to detect ischemia area and hypertrophy in histology, which will support the basis of the whole studies.
  • Isolated muscle contraction may not represent the cardiac phenotype in vivo, therefore, any indications of HF would be helpful to backup the claim of “compensation” vs. “decompensation” state of the hearts.
  • The variabilities of MLCK, ZIPK and MYPT2 in sham group over time are substantial. Therefore, further prove of the relationship between those enzymes and RLC or MYBPC phosphorylations would be required.    

Reviewer 2 Report

The authors aimed to examine a detailed characterization of the physiological and biochemical changes in vivo after heart failure and  succeeded to relate phosphorylation level of regulatory light chains (RLC) to force production.      They selected the RLC phosphorylation from many other proteins. The experiments were well conducted.  However, I raised a few questions and comments.  

This is only a serious problem. The author measured tension of Triton-skinned fiber. During over-night skinning, phosphorylation levels may be changed. Please comment on the text.

Reactive oxygen is produced by MI.  Oxidation of proteins, enzyme, RLC etc. may modify the function under MI. Please comment on the text.

They measured enzyme expression. Is it possible to measure enzyme activity? They used SDS-PAGE for quantification of phosphorylation.   Why not use mass-spectroscopy?

ATPase rate and ATP level are also important because ATP is depleted under myocardial infarction (MI).  

Lower ATPase rate and higher force must be achieved economically with high efficiency. Why not measure ATPase? Please comment in the text. 

The authors emphasized that RLC phosphorylation is important because it produces higher force. However, it is unclear why higher force is important. Force production is high in compensentary phase or low in decompensatory phase.  Why is this benefits for failing heart?  Please comment in the text.

Round 2

Reviewer 1 Report

The reply to the comments are reasonable and the additional information can improve the manuscript.